# RAFT: REASONING-AWARE FINE-TUNING

## ABSTRACT

Supervised fine-tuning (SFT) adapts large pretrained models to downstream tasks but often fails to learn reasoning-consistent mappings, especially under limited data. Chain-of-Thought (CoT) fine-tuning addresses this by training models to produce explicit reasoning traces, but at the cost of significantly increased inference latency and variable effectiveness across domains. We introduce *Reasoning-Aware Fine-Tuning (RAFT)*, a single-stage framework that distils reasoning signals during training without requiring reasoning generation at inference. RAFT leverages a reasoning-discriminative loss applied to positive and negative reasoning traces sampled from a teacher model, guiding the student to align its internal scoring with valid reasoning while preserving the efficiency of SFT. Our extensive experiments across visual reasoning, medical VQA, fine-grained recognition, and CommonsenseQA demonstrate that RAFT consistently outperforms SFT and CoT-FT baselines, while maintaining SFT-level inference efficiency. Beyond accuracy, we provide the systematic analysis of RAFT's *scalability and robustness*: (i) performance improves monotonically with stronger teachers (3B–GPT-4.1), and (ii) RAFT remains effective even with noisy teacher supervision. Compared against preference-optimisation baselines, RAFT delivers complementary advantages by distilling reasoning rather than preferences.

## 1 INTRODUCTION

The advent of large pretrained models such as BERT (Devlin et al., 2019), GPT-2 (Radford et al., 2019), and GPT-3 (Brown et al., 2020) has revolutionised the field of artificial intelligence, offering unprecedented capabilities across a wide range of tasks. A key paradigm for leveraging their power for specific applications is Supervised Fine-Tuning (SFT) (Wei et al., 2021; Ouyang et al., 2022). SFT adapts these general-purpose models by further training them on smaller, task-specific datasets. This process updates the model's parameters, enabling it to handle a broader range of tasks.

Although widely used, standard SFT is not without its limitations. Our empirical analysis in section 4.1 reveals that, when trained on limited non-reasoning data, SFT tends to learn *surface-level* correlations sufficient for producing correct answers, but does not capture the reasoning process underlying them, under limited supervision. For instance, as shown in Figure 1, an SFT model may produce the correct answer, but when asked to discriminate between correct and incorrect reasoning steps, it tends to prefer the incorrect one. This suggests that the model has not learned to ground its output in valid reasoning, but rather relies on spurious input-output patterns.

To address the limitations of SFT in capturing explicit input-output mapping, Chain-of-Thought (CoT) fine-tuning (Kim et al., 2023; Muennighoff et al., 2025; Ye et al., 2025) has gained traction. This approach mitigates the shortcomings of SFT by explicitly encouraging models to generate intermediate reasoning steps that lead to the final answer. By making the reasoning process explicit, CoT fine-tuning can significantly improve performance on tasks demanding complex, multi-step deduction (Muennighoff et al., 2025; Ye et al., 2025). However, this improvement comes at a cost. The generation of explicit reasoning inevitably increases the length of output sequences, thereby introducing a higher per-example inference latency as shown in Figure 5c. This can be a critical bottleneck for applications with strict online latency requirements. Furthermore, a recent study (Li et al., 2025) and our investigations reveal that CoT fine-tuning, while beneficial for complex tasks, could potentially diminish performance on tasks that do not *necessitate* such complex reasoning (*e.g.*, visual recognition in Table 2), especially without sufficient high-quality reasoning data.

Figure 1: Illustration of the limitation of SFT in low-data regimes. Given a visual question, the SFT model trained on "prompt-answer" pairs correctly prefers the right answer with a high probability. However, the model shows a low preference for the correct reasoning. This suggests that such an SFT model can answer correctly without grasping the underlying valid reasoning process.

Therefore, we propose a novel fine-tuning pipeline **Reasoning-Aware Fine-Tuning (RAFT)**. The core of RAFT lies in its unique integration of reasoning supervision directly into the fine-tuning objective. This is achieved via a *reasoning-discriminative loss*, which explicitly guides the model to favour correct reasoning tokens over incorrect ones. The supervision for this discriminative process is derived from contrasting positive (correct) and negative (incorrect) reasoning examples, often sampled with the aid of a more capable teacher model. Crucially, this supervision is merely applied during training without necessitating the explicit generation of these reasoning tokens at inference time. Consequently, RAFT aims to preserve the computational efficiency characteristic of standard SFT while simultaneously enhancing the model's ability to internally differentiate between valid and invalid reasoning. RAFT achieves significant accuracy improvements in visual reasoning and fine-grained recognition, demonstrating a highly effective and efficient approach to adapting models with an awareness of underlying reasoning discrimination.

The primary contributions of this work are: (i) Our analysis reveals that standard SFT, trained only on final answers, largely fails to distinguish between valid and flawed reasoning tokens. This highlights the critical need for RAFT's explicit reasoning-aware supervision. (ii) *A general reasoning-distillation framework*: We propose RAFT, a single-stage fine-tuning method that introduces a reasoning-discriminative loss, enabling models to internalise reasoning signals without inference–time overhead. (iii) *Comprehensive empirical validation*: We show RAFT consistently improves performance across visual reasoning, fine-grained recognition, and text-only tasks, outperforming both SFT and CoT-FT under few-shot and low-resource conditions. (iv) *Scalability and robustness analysis*: We introduce a new metric, *Teacher CoT Accuracy*, to systematically measure the quality of reasoning supervision and demonstrate monotonic student performance improvements with stronger teachers (from 3B to GPT-4.1). We analyse the cost-performance trade-off, showing RAFT remains effective with reduced negative samples and maintains robustness under noisy supervision.

## 2 RELATED WORK

**Fine-tuning techniques.** Fine-tuning is the process of customising a large pre-trained model (Radford et al., 2019; Brown et al., 2020) to perform optimally on a specific task by further training it on a smaller, task-specific dataset. This process updates the model's parameters to help it acquire new abilities, whereas in-context learning (Brown et al., 2020) leaves the model's parameters unchanged. *SFT* (Wei et al., 2021; Ouyang et al., 2022; Liu et al., 2023)—where the model learns from example responses by maximising the likelihood. An earlier study, FLAN (Wei et al., 2021), shows that fine-tuning with a wide range of instruction-based datasets greatly improves zero-shot performance on tasks the model has not encountered before. *Direct Preference Optimisation (DPO)* (Rafailov et al., 2023)—where the model is shown both preferred and non-preferred responses to better capture subjective preferences; and *Reinforcement Fine-Tuning (RFT)* (Schulman et al., 2017; Jaech et al., 2024; Guo et al., 2025)—where the reinforcement learning technique is used to enhance the models' reasoning capabilities in reasoning tasks such as solving mathematical problems (Luong et al., 2024; Shao et al., 2024b; Yang et al., 2024) and coding (Hui et al., 2024; Zhang et al., 2024b).

**SFT with reasoning data.** Recent research has emphasised the value of fine-tuning pre-trained models using datasets that include detailed reasoning steps (Muennighoff et al., 2025; Ye et al., 2025; Li et al., 2025). For example, Muennighoff et al. (2025) curated a high-quality dataset of 1000 question-reasoning pairs, carefully chosen for their challenge, diversity, and reasoning clarity. Fine-tuning the Qwen2.5 32B-Instruct model on this data led to performance gains of up to 27%

over o1-preview on competitive math questions. Similarly, LIMO (Ye et al., 2025) achieved 57.1% accuracy on the demanding AIME benchmark by training on just 817 curated samples. However, a recent study (Li et al., 2025) indicates that not all the tasks can benefit from fine-tuning with reasoning data. Thus, the goal of our work is to explore whether we can leverage reasoning data to improve task performance that seemingly does not require explicit reasoning outputs.

**Parameter-efficient fine-tuning (PEFT).** As state-of-the-art pre-trained models, such as GPT-4o (Hurst et al., 2024), continue to grow in size and complexity, fully fine-tuning all their parameters becomes increasingly resource-intensive and expensive. To address this challenge, recent research has introduced PEFT strategies (Lester et al., 2021; Hu et al., 2022; Zhang et al., 2023a), which focus on updating only a small subset of the model's weights or incorporating lightweight modules such as adapters (Zhang et al., 2023a), prompt tuning techniques (Lester et al., 2021), or low-rank adaptation methods like LoRA (Hu et al., 2022). These approaches represent a significant advancement in the field, as they enable practitioners to adapt large models to new tasks with substantially reduced computational and memory requirements.

**Preference Optimisation.** Preference optimisation methods such as DPO (Rafailov et al., 2023), ORPO (Hong et al., 2024), and GRPO (Shao et al., 2024b) have recently emerged as powerful approaches for aligning large language models with human or synthetic preferences. These methods train models to prefer desirable responses over undesirable ones by directly optimising preference data. While highly effective for subjective alignment tasks (e.g., dialogue quality or safety), their supervision signal focuses on end-level preferences rather than the intermediate reasoning process. In contrast, RAFT complements these methods by explicitly distilling *reasoning signals*. Rather than optimising for user preference, RAFT employs a reasoning-discriminative loss over positive and negative reasoning traces, teaching the model to internally favour valid reasoning steps without forcing reasoning generation at inference time. Thus, RAFT can be viewed as a reasoning-centric counterpart to preference optimisation, offering orthogonal benefits in reasoning-intensive domains. Detailed distinction with DPO can be found in section A.9.

**Scaling Laws of Chain-of-Thought.** Recent work has studied the scaling behaviour of Chain-of-Thought (CoT) fine-tuning. Muennighoff et al. (2025) demonstrate that CoT benefits scale significantly with model size and data quality, but improvements are concentrated in domains requiring complex symbolic reasoning, such as mathematics and coding. Similarly, Sprague et al. (2024) shows that the effectiveness of CoT is highly domain-dependent, with minimal or even negative gains on recognition-heavy or low-data tasks. These findings highlight an important limitation: while CoT excels in math/code settings, its efficiency and generalisation can degrade in broader applications. RAFT addresses this gap by *leveraging reasoning supervision without inference-time CoT generation*, allowing reasoning-aware improvements even in non-symbolic or low-resource domains where CoT may struggle.

## 3 PRELIMINARIES

**Notation.** In this section, we introduce the mathematical notation in this paper. Let $\mathcal{X}$ denote the set of prompts and $\mathcal{V}$ denote the set of vocabulary. The policy model[1] $\pi_\theta$ parameterised by $\theta$ outputs a probability distribution $\pi_\theta(\mathbf{y}|\mathbf{x})$, where $\mathbf{x} = [x_1, \ldots, x_N] \in \mathcal{X}$ is the sequence of input tokens and $\mathbf{y} = [y_1, \ldots, y_L] \in \mathcal{V}^L$ is the sequence of output tokens. Typically, the policy model $\pi_\theta$ is an auto-regressive model (Wang et al., 2024; Bai et al., 2025), meaning that it predicts the output probability of the $y_l$ given all tokens in $\mathbf{x}$ and $\mathbf{y}_{<l}$ as follows:

$$\pi_\theta(\mathbf{y}|\mathbf{x}) := \prod_{l=1}^{L} \pi(y_l|\mathbf{x}, \mathbf{y}_{<l}), \tag{1}$$

where $\mathbf{y}_{<l} := [y_1, \ldots, y_{l-1}]$ and $\mathbf{y}_{<1}$ is null.

**Problem setting.** The fundamental task addressed in this work is the adaptation of $\pi_\theta$ for enhanced performance on a downstream task given its training dataset $\mathcal{D} = \{(\mathbf{x}_i, \mathbf{y}_i)\}_{i=1}^{N}$, where $\mathbf{x}_i \in \mathcal{X}$ represents input tokens and $\mathbf{y}_i \in \mathcal{V}^*$ is the corresponding output tokens. Starting from a pre-trained state, the model $\pi_\theta$ is optimised to effectively map inputs $\mathbf{x}$ characteristic of the downstream task to their target outputs $\mathbf{y}$, typically by optimising an objective function derived from $\mathcal{D}$.

---

[1]The model we want to fine-tune.

**SFT.** SFT (Wei et al., 2021; Liu et al., 2023) is a standard approach to adapt $\pi_\theta$ to a specific task by directly maximising the likelihood of target outputs. Formally, given a dataset $\mathcal{D}$, the training objective is to minimise the negative log-likelihood of the outputs conditioned on the input tokens:

$$\min_\theta \mathcal{J}_{\text{SFT}}(\theta) := -\mathbb{E}_{(\mathbf{x},\mathbf{y})\sim\mathcal{D}}[\log \pi_\theta(\mathbf{y} \mid \mathbf{x})]. \tag{2}$$

**CoT fine-tuning.** CoT fine-tuning (Kim et al., 2023; Muennighoff et al., 2025; Ye et al., 2025) encourages $\pi_\theta$ to produce the intermediate reasoning processes, improving performance on complex reasoning tasks. Specifically, given a dataset $\mathcal{D}_{\text{CoT}} = \{(\mathbf{x}_i, \mathbf{z}_i)\}_{i=1}^N$ where $\mathbf{z}_i = (\mathbf{r}_i \oplus \mathbf{y}_i)$ denotes the concatenation of the reasoning tokens $\mathbf{r}$ and output answer tokens $\mathbf{y}$. CoT fine-tuning maximises the likelihood of generating both the reasoning and the final answer tokens. Formally, the objective is

$$\min_\theta \mathcal{J}_{\text{CoT}}(\theta) := -\mathbb{E}_{(\mathbf{x},\mathbf{z})\sim\mathcal{D}_{\text{CoT}}} \left[\log \pi_\theta(\mathbf{z} \mid \mathbf{x})\right]. \tag{3}$$

By training the model to predict not only the final answer but also the reasoning tokens, this enables $\pi_\theta$ to have better performance on those complex tasks (Kim et al., 2023; Shao et al., 2024a). However, producing both reasoning and answer typically increases the length of each generated sequence, which in turn raises *per-example inference time*. This might pose challenges for applications with strict online inference latency requirements.

## 4 REASONING-AWARE FINE-TUNING

In this section, we conduct a preliminary step to show that standard SFT largely fails to discriminate between correct and incorrect reasoning, even though the model can output the correct answer directly. Then we elaborate on the proposed method, RAFT.

### 4.1 ANALYSING THE ABILITY TO DISCRIMINATE REASONING DURING SFT

We investigate whether SFT (trained on $(\mathbf{x}, \mathbf{y})$ pairs) enables the model $\pi_\theta$ to implicitly learn the underlying reasoning $\mathbf{r}$ from a prompt $\mathbf{x}$ to its corresponding answer $\mathbf{y}$. Specifically, we aim to determine if models fine-tuned with SFT develop an ability to recognise valid reasoning steps over incorrect ones. Understanding this is crucial: if SFT models naturally learn to value correct reasoning, our task might merely involve amplifying this existing signal. However, if they do not, then a more direct intervention to teach $\pi_\theta$ for reasoning discrimination during training is warranted. In this section, we empirically investigate this research question:

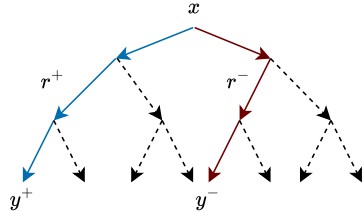

Figure 2: Illustrating the reasoning sampling protocol.

> *Does SFT of $\pi_\theta$ on $\mathcal{D}$ alone enable the model to implicitly learn to distinguish valid reasoning pathways that map a prompt $\mathbf{x}$ to its correct answer $\mathbf{y}$?*

To assess whether SFT implicitly learn the reasoning discriminative capability, we must first address a problem: standard SFT datasets $\mathcal{D} = \{(\mathbf{x}_i, \mathbf{y}_i)\}_{i=1}^N$ lack explicit reasoning annotations. Thus, we propose a *reasoning sampling protocol* to annotate the reasoning from a teacher model.

#### 4.1.1 REASONING SAMPLING PROTOCOL

**Reasoning sampling.** Here, we synthesise reasoning steps using a teacher model $\pi^\mathcal{T}$. Even though $\mathcal{D}$ lacks reasoning annotations, a teacher model (*e.g.*, a larger pretrained model or human annotator) can generate plausible reasoning $\mathbf{r}_i^+$ that leads to $\mathbf{y}_i$ and contrastive "distract" reasoning $\mathbf{r}_i^-$ that yields incorrect answers $\mathbf{y}_i^- \neq \mathbf{y}_i$. For each $(\mathbf{x}_i, \mathbf{y}_i)$, we generate contrastive reasoning tokens using a teacher model $\pi^\mathcal{T}$. For each prompt $x_i$, we sample $k = 50$ completions from $\pi^\mathcal{T}$ with temperature $T = 1$ and top-$p = 1$:

$$\{\mathbf{z}_i^{(1)}, \dots, \mathbf{z}_i^{(k)}\} \sim \pi^\mathcal{T}\left(\cdot \mid \tilde{\mathbf{x}}_i\right), \tag{4}$$

where $\tilde{\mathbf{x}}_i =$ "First reason step-by-step, then answer: " $\oplus \mathbf{x}_i$. Among these completions, we select:

$$\mathbf{z}_i^+ \sim \text{Uniform}\{ \mathbf{z}_i^{(m)} : \mathbf{y}^{(m)} = \mathbf{y}_i \}, \quad \mathbf{z}_i^- \sim \text{Uniform}\{ \mathbf{z}_i^{(m)} : \mathbf{y}^{(m)} \neq \mathbf{y}_i \}, \tag{5}$$

where $\mathbf{z}_i^+ = \mathbf{r}_i^+ \oplus \mathbf{y}_i^+$, $\mathbf{z}_i^- = \mathbf{r}_i^- \oplus \mathbf{y}_i^-$. To further clarify our reasoning-trace construction protocol, we provide concrete worked examples in Appendix A.10. This ensures each training instance $(x_i, z_i^+, z_i^-)$ contains at least one valid positive and one valid negative reasoning path. If no valid pair is found, we discard the instance for that epoch, preserving stability.

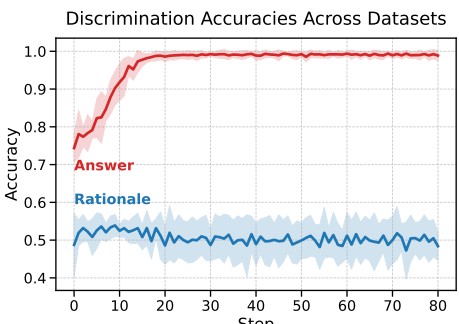

Discrimination Accuracies Across Datasets

Figure 3: Discrimination acc.

For the positive response $\mathbf{r}_i^+ \oplus \mathbf{y}_i^+$, where $\mathbf{r}_i^+$ is referred to as positive reasoning tokens, as its corresponding answer $\mathbf{y}_i^+$ matches the ground truth $\mathbf{y}_i$. For a negative response $\mathbf{r}_i^- \oplus \mathbf{y}_i^-$, which leads to a wrong answer $\mathbf{y}_i^-$. Thus, we can construct a dataset $\mathcal{D}^{\text{RAFT}} = \{\mathbf{x}_i, \mathbf{z}_i^+, \mathbf{z}_i^-\}_{i=1}^N$. This protocol ensures that negatives are not trivial noise but represent *reasonable yet flawed reasoning*, providing a contrastive supervision signal that helps the student discriminate between correct and incorrect reasoning paths.

**Robustness to noisy reasoning.** In practice, teacher generations may contain ambiguous or noisy reasoning traces. To study RAFT's robustness, we perform ablations where negative pairs are sampled from partially correct but logically inconsistent reasoning. Results (see section 5) show RAFT remains stable under such noise, demonstrating that its discriminative signal is resilient to imperfect supervision.

### 4.1.2 REASONING DISCRIMINATION ACCURACY

To evaluate the ability of a model to distinguish between valid and invalid reasoning paths, we define a unified metric: **Reasoning Discrimination Accuracy (RDA)**. Given a prompt $x_i$ and a pair of reasoning traces $\mathbf{z}_i^+ = \mathbf{r}_i^+ \oplus \mathbf{y}_i^+$ (correct) and $\mathbf{z}_i^- = \mathbf{r}_i^- \oplus \mathbf{y}_i^-$ (incorrect), we compute

$$\text{RDA}(\pi) = \frac{1}{N} \sum_{i=1}^N \mathbb{I}\Big[\tilde{\pi}(\mathbf{z}_i^+ \mid \mathbf{x}_i) > \tilde{\pi}(\mathbf{z}_i^- \mid \mathbf{x}_i)\Big], \tag{6}$$

where $\tilde{\pi}(\mathbf{z} \mid \mathbf{x})$ denotes the length-normalised likelihood (Meng et al., 2024) of sequence $z$ under model $\pi$. A reasoning trace is considered positive only if its *final prediction matches the ground truth*; Otherwise it is negative, regardless of fluency or surface probability. The likelihood is therefore not treated as an absolute proxy for reasoning quality, but only as a relative scoring signal: the model is encouraged to rank verified-positive traces above verified-negative ones. This formulation is consistent with preference-optimisation and reinforcement-style methods such as GRPO (Shao et al., 2024b) and DPO (Rafailov et al., 2023), which also rely on relative likelihood comparisons between desirable and undesirable responses. RDA can be applied to different models: (i) When $\pi = \pi_T$ (teacher), RDA measures the reliability of teacher supervision (*Teacher CoT Accuracy*). (ii) When $\pi = \pi_\theta$ (student), RDA measures how well the student has internalized reasoning discrimination (*Student Discrimination Accuracy*).

**Empirical results.** To examine what SFT learns, we directly evaluate the discriminative ability of the model on its *training set* using the RDA metric (section 4.1.2). For each training instance, we test whether the SFT model $\pi_\theta$[2] assigns higher likelihood to the ground-truth-verified reasoning trace $\mathbf{z}^+$ than to the incorrect reasoning trace $\mathbf{z}^-$. As shown in Figure 3, while SFT substantially improves *answer discrimination*, its *reasoning discrimination* on the training set remains near chance. This indicates that SFT effectively learns input–output mappings but fails to acquire the ability to distinguish between $\mathbf{r}^+$ and $\mathbf{r}^-$. The near-chance training-set discrimination performance demonstrates the necessity of explicit supervision for reasoning discrimination, which motivates RAFT's reasoning-discriminative objective.

---

[2] The SFT model $\pi_\theta$ is trained *only* on $\mathcal{D}$ and does not observe $(\mathbf{z}_i^+, \mathbf{z}_i^-)$ pairs or the teacher model $\pi^{\mathcal{T}}$ during its SFT phase.

Figure 4: Overview of the RAFT pipeline. Given an input prompt, a refined prompt $\tilde{\mathbf{x}}$ is appended to the input prompt, then passed to a teacher model to sample a pair of reasoning steps: a positive reasoning leading to the correct answer, and a negative reasoning leading to an incorrect one. The model is then fine-tuned to maximise the likelihood of the correct answer while learning to prefer the positive reasoning over the negative one using a reasoning discriminative objective.

## 4.2 OBJECTIVE FOR REASONING-AWARE FINE-TUNING

A principled way to achieve this is to adopt the Bradley–Terry (BT) model (Bradley & Terry, 1952), a widely used probabilistic framework for modelling pairwise preferences. The BT model defines the probability that one candidate $\mathbf{y}_1$ is preferred over another $\mathbf{y}_2$ given an input $\mathbf{x}$ as:

$$\mathbb{P}_{\mathrm{BT}}(\mathbf{y}_1 \succ \mathbf{y}_2 \mid \mathbf{x}) = \sigma\big(r(\mathbf{y}_1) - r(\mathbf{y}_2)\big), \tag{7}$$

where $r(\cdot)$ is a scalar function that represents the model's internal preference, and $\sigma(\cdot)$ is the sigmoid function. This formulation captures the intuition that the larger the margin between $\mathbf{y}_1$ and $\mathbf{y}_2$, the more confident we are that $\mathbf{y}_1$ is the preferred response. By incorporating this pairwise comparison into the training objective, we can directly supervise the model to assign higher likelihood to positive reasoning, thereby aligning its internal scoring function with the quality of reasoning paths, even when the model is trained only to produce final answers at inference time. We can come up with the following objective

$$-\mathbb{E}_{(\mathbf{x},\mathbf{y},\mathbf{z}^+,\mathbf{z}^-)\sim\mathcal{D}^{\mathrm{RAFT}}}\left[\underbrace{\log \pi_\theta(\mathbf{y} \mid \mathbf{x})}_{\text{SFT loss}} + \underbrace{\beta \log \sigma\big(r(\mathbf{x},\mathbf{z}^+) - r(\mathbf{x},\mathbf{z}^-)\big)}_{\text{Reasoning discriminative loss}}\right], \tag{8}$$

where $\beta$ is a weighting hyperparameter, and the reasoning discriminative loss term could be realised by the log of odds (Hong et al., 2024) as

$$\mathrm{logit}(\mathbf{y} \mid \mathbf{x}) = \log \frac{\tilde{\pi}_\theta(\mathbf{y} \mid \mathbf{x})}{1 - \tilde{\pi}_\theta(\mathbf{y}|\mathbf{x})}, \tag{9}$$

where $\frac{\tilde{\pi}_\theta(\mathbf{y}|\mathbf{x})}{1-\tilde{\pi}_\theta(\mathbf{y}|\mathbf{x})} = k$ indicates that it is $k$ times more likely for the model $\tilde{\pi}_\theta$ to generate the output sequence $\mathbf{y}$ than not generating it. Also, $\sigma(r(\mathbf{x},\mathbf{z}^+) - r(\mathbf{x},\mathbf{z}^-))$ defines an odds ratio to indicate how likely it is for the model to generate $\mathbf{z}^+$ than $\mathbf{z}^-$.

**Prompt perturbation.** To prevent the model from conditioning all three sequences $\mathbf{y}$, $\mathbf{z}^+$ and $\mathbf{z}^-$ on the identical prompt $\mathbf{x}$ in Equation 8, we slightly modify $\mathbf{x}$ in the reasoning discriminative term as $\tilde{\mathbf{x}}$. We apply this prompt perturbation to disambiguate the SFT and reasoning discriminative loss, enabling the model to distinguish between imitation and comparison signals during training, while still supporting efficient, reasoning-free decoding at inference time. Thus, our final objective is

$$\mathcal{J}_{\mathrm{RAFT}}(\theta) := -\mathbb{E}_{(\mathbf{x},\mathbf{y},\mathbf{z}^+,\mathbf{z}^-)\sim\mathcal{D}^{\mathrm{RAFT}}}\left[\log \pi_\theta(\mathbf{y}|\mathbf{x}) + \beta \log \sigma\big(\mathrm{logit}(\mathbf{z}^+|\tilde{\mathbf{x}}) - \mathrm{logit}(\mathbf{z}^-|\tilde{\mathbf{x}})\big)\right], \tag{10}$$

**RAFT Outline.** The RAFT pipeline (Figure 4) is as follows: given a dataset $\mathcal{D}$, for each $(\mathbf{x}_i, \mathbf{y}_i)$ we invoke a fixed teacher model $\pi^{\mathcal{T}}$ to sample $(\mathbf{z}_i^+, \mathbf{z}_i^-)$ under a modified prompt $\tilde{\mathbf{x}}_i$. we fine-tune our policy model $\pi_\theta$ by minimising the RAFT loss in Equation 10. Finally, at deployment

Table 1: Performance comparison of different methods on Visual Reasoning and Visual Medical Reasoning benchmarks. Scores are reported as accuracy (%). The base model is Qwen2.5-VL-7B.

| Method | Visual Reasoning | Visual Medical Reasoning | | Avg. |
|---|---|---|---|---|
| | CVQA | OmniMedVQA | PMC-VQA | |
| Qwen2.5-VL-7B (Bai et al., 2025) | 51.81 | 65.00 | 48.00 | 54.94 |
| Qwen2.5-VL-72B (Bai et al., 2025) | 71.08 | 67.38 | 55.60 | 64.68 |
| SFT (Wei et al., 2021) | 58.46 | 81.50 | 49.80 | 63.25 |
| COT-FT (Kim et al., 2023) | 58.00 | 83.13 | 46.60 | 62.58 |
| RAFT (*Ours*) | **64.47 ± 0.04** | **87.63 ± 0.10** | **53.53 ± 0.07** | **68.54** |

Table 2: Performance comparison of different methods on Fine-grained Visual Recognition (few-shot) benchmarks. Scores reported as Accuracy (%). The base model is Qwen2.5-VL-7B. RAFT shows mean ± std over three runs (std on a second line).

| Dataset | Zero-shot | | 1-shot | | | 4-shot | | | 8-shot | | |
|---|---|---|---|---|---|---|---|---|---|---|---|
| | Qwen2.5-VL-7B | Qwen2.5-VL-72B | SFT | CoT-FT | **RAFT** | SFT | CoT-FT | **RAFT** | SFT | CoT-FT | **RAFT** |
| CUB | 37.40 | 64.80 | 66.60 | 43.70 | **71.50** ±0.05 | 79.70 | 60.70 | **83.10** ±0.04 | 83.10 | 58.90 | **84.80** ±0.06 |
| S.Dogs | 12.50 | 65.83 | 78.50 | 57.50 | **80.30** ±0.07 | 81.83 | 60.50 | **82.50** ±0.05 | 81.67 | 64.67 | **83.83** ±0.09 |
| F.Aircraft | 42.75 | 62.00 | 70.57 | 46.00 | **73.71** ±0.06 | 79.14 | 61.71 | **81.14** ±0.08 | 81.43 | 65.43 | **81.43** ±0.07 |
| O.Pet | 64.43 | 78.63 | 82.16 | 49.22 | **84.32** ±0.04 | 81.62 | 72.97 | **85.41** ±0.05 | 83.78 | 75.14 | **84.86** ±0.06 |
| S.Cars | 35.77 | 75.71 | 73.47 | 37.96 | **74.29** ±0.05 | 82.14 | 60.51 | **83.98** ±0.07 | 85.41 | 70.20 | **87.14** ±0.06 |
| Avg. | 38.57 | 69.40 | 74.26 | 46.87 | **76.28** | 80.89 | 63.28 | **83.23** | 83.08 | 66.87 | **84.41** |

time, we condition only on $\mathbf{x}_{\text{test}}$ so that the model produces the answer directly, omitting reasoning generation and thus preserving low-latency inference while retaining the benefits of reasoning-aware supervision. The overall algorithm can be found in the Appendix.

## 5 EXPERIMENTS

To empirically validate the effectiveness and efficiency of our proposed method RAFT, we conducted comprehensive experiments across diverse benchmark datasets.

**Implementation details.** Datasets and baselines can be found in Appendix A.1. During the training in all experiments, we use the Qwen2.5-VL-7B (Bai et al., 2025) vision-language model as $\pi_\theta$, leveraging the PyTorch (Paszke, 2019) framework and the Hugging Face Transformers library (Wolf et al., 2019). Fine-tuning was performed on 8 NVIDIA H100 GPUs. We use the AdamW optimiser with a learning rate of 1e-4 and a cosine decay schedule (Loshchilov & Hutter, 2016) with a warm-up ratio of 0.05, using a global batch size of 32. Training proceeds for 400-1600 steps based on the size of the dataset. All the methods are fine-tuned with LoRA (Hu et al., 2022) with rank 4, alpha as 8. For RAFT, the teacher model $\pi^{\mathcal{T}}$ used for sampling is Qwen2.5-VL-72B (Bai et al., 2025). The reasoning discriminative object weight $\beta$ was set to 0.001, and we appended "First reason step-by-step, then answer" to the input prompt. Baselines are implemented following standard procedures; specifically, SFT and COT-FT use standard cross-entropy loss. Evaluation metrics follow common practice for each benchmark dataset.

**Visual reasoning results.** Table 1 summarises the performance comparison of RAFT against baseline methods on Visual Reasoning and Visual Medical Reasoning benchmarks, reporting accuracy (%). First, we can observe that integrating CoT with SFT leads to a discrepant effect, *e.g.*, +1.6% on OminiMedVQA while -3.2% on PMC-VQA. This indicates that naively urging CoT output tokens does not always facilitate adaptation with downstream tasks. In contrast, RAFT demonstrates superior performance across all tasks, *i,e,*, 65.08% on CVQA, 88.75% on OmniMedVQA, and 53.60% on PMC-VQA, providing its advantage in utilising reasoning. Second, compared with SFT, the proposed method attains notable performance gain, without sacrificing on inference efficiency (see analysis in Figure 5c). These results strongly validate the efficacy of the RAFT approach.

Table 3: Performance comparison of RAFT and DPO-based methods across reasoning and fine-grained recognition benchmarks. Scores are reported as accuracy (%).

| Method | CVQA | OmniMed | PMC | CUB | S.Dogs | F.Aircraft | Pet | S.Cars | Avg. |
|---|---|---|---|---|---|---|---|---|---|
| DPO (only) | 45.08 | 60.88 | 50.60 | 47.40 | 67.50 | 52.00 | 75.68 | 47.65 | 55.85 |
| DPO (2-stage) | 58.62 | 86.38 | 50.20 | 67.90 | 79.17 | 69.43 | 80.00 | 73.06 | 70.60 |
| SimPO (2-stage) | 58.77 | 87.00 | 50.60 | 67.60 | 79.17 | 69.14 | 78.92 | 74.39 | 70.70 |
| ORPO (2-stage) | 59.23 | 86.88 | 51.00 | 68.40 | 77.00 | 68.29 | 80.54 | 73.57 | 70.61 |
| RAFT (*Ours*) | **65.08** | **88.75** | **53.60** | **71.50** | **80.17** | **73.71** | **83.24** | **74.29** | **73.79** |

**Fine-grained visual recognition results.** To better validate the effectiveness of RAFT, we evaluated its performance on Fine-grained Visual Recognition tasks in a few-shot setting, with results presented in Table 2. First, our method consistently yields improvement over all datasets with varying shots, which generally validates the effectiveness of the approach. Second, a noticeable phenomenon is that CoT-FT leads to inferior performance on all scenarios. We conjecture that, under limited data conditions, CoT-FT struggles to fully capture and internalise the logic expressed in CoT reasoning. This suggests that enforcing CoT output is not always necessary or beneficial, particularly in few-shot scenarios. However, as the amount of CoT-supervised data increases, we observe a corresponding improvement in the performance of CoT fine-tuning. Nevertheless, our method can still use this reasoning data to improve performance, which reflects the effectiveness of our method.

**Comparison with DPO-based Methods.** To ensure a fair comparison between RAFT and DPO-style preference optimization, we carefully mirrored RAFT's joint objective structure in our baselines. Specifically, for DPO we adopted a two-stage pipeline: (1) supervised fine-tuning (SFT) on ground-truth labels to provide task supervision, followed by (2) preference optimisation over reasoning traces. This setup ensures that both RAFT and DPO variants are exposed to equivalent task-level information and differ only in how preference signals are incorporated.

Empirical results are reported in Table 3. RAFT consistently outperforms both vanilla DPO and stronger two-stage variants (SimPO, ORPO) across reasoning (CVQA, OmniMed, PMC-VQA) and fine-grained recognition benchmarks (CUB, Stanford-Dogs, FGVC-Aircraft, Pet, Stanford-Cars), achieving an average accuracy of **73.79%**, compared to ∼70.6–70.7% for the strongest DPO-based baselines. These results validate that RAFT's integrated supervision over both answers and reasoning traces yields stronger generalisation and higher task accuracy than DPO-style alternatives.

**Impact of Reasoning Discrimination Weight** $\beta$**.** We analyse the sensitivity of RAFT to the hyperparameter $\beta$, which balances the SFT loss with the reasoning-discriminative loss. The results, presented in Figure 5, illustrate the dual impact of $\beta$: its effect on the final downstream task accuracy and its influence on the model's ability to learn discriminative reasoning. Figure 5a shows the final accuracy achieved on a representative task across a range of $\beta$ values, from $10^{-1}$ down to $10^{-4}$. We observe that an intermediate value, specifically $\beta = 10^{-3}$, yields the optimal performance. This suggests that while reasoning supervision is beneficial, an excessively large $\beta$ might overshadow the primary task objective, or too small a $\beta$ might not provide sufficient signal for learning robust reasoning. Concurrently, Figure 5b details the training dynamics of the model's CoT accuracy, which measures its proficiency in distinguishing correct from incorrect reasoning tokens. It is evident that larger values of $\beta$ (e.g., $\beta = 1.0$ and $\beta = 0.1$) accelerate the learning of this discriminative capability, leading to faster convergence and higher saturation levels in CoT accuracy. This confirms that the reasoning discriminative loss is able to help distinguish the correct and incorrect reasoning steps.

**Token Efficiency.** One of the core advantages of RAFT lies in its ability to incorporate reasoning supervision during training while maintaining inference-time efficiency. Figure 5c compares the average number of output tokens produced by SFT, CoT-FT, and RAFT across six fine-grained visual recognition benchmarks. As shown, CoT-FT significantly increases the output length, often requiring over 200 tokens per prediction due to its need to generate detailed reasoning. In contrast, both SFT and RAFT maintain concise outputs (typically under 15 tokens), making them more suitable for latency-sensitive applications. Notably, RAFT matches or exceeds the performance of CoT-FT while retaining the token efficiency of SFT, thereby offering a favourable trade-off between reasoning ability and computational overhead.

**Influence of** $\pi^{\mathcal{T}}$ **Scale.** We conduct a systematic scaling study using Qwen2.5-VL teachers of varying sizes (3B, 7B, 72B) and a GPT-4.1 teacher against a Zero-Shot baseline. As Figure 6 shows that RAFT's performance improves *monotonically* with stronger teachers across both reasoning and

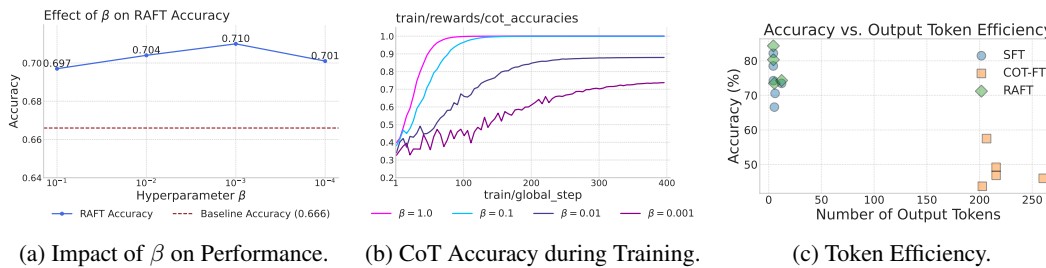

(a) Impact of $\beta$ on Performance.     (b) CoT Accuracy during Training.     (c) Token Efficiency.

Figure 5: Detailed empirical analysis on fine-grained visual recognition tasks.

recognition benchmarks. This validates our theoretical motivation: the reasoning-discriminative loss transfers stronger reasoning signals when the teacher's CoT accuracy is higher. Importantly, RAFT consistently improves over its SFT baseline even with small teachers (3B), demonstrating that RAFT is effective without requiring an extremely strong teacher.

**Sensitivity to Teacher Bias.** A natural concern is whether RAFT merely inherits the biases of the teacher model, rather than learning to prefer correct reasoning in its own right. In Figure 6: RAFT *remains effective even with weaker teachers* (baseline). This suggests that RAFT does not simply imitate teacher biases, but rather extracts useful contrastive reasoning signals across a range of teacher qualities. RAFT is explicitly designed to reduce sensitivity to individual teacher idiosyncrasies through two mechanisms: (1) **Pairwise, margin-based reasoning supervision.** RAFT does not directly imitate a single teacher's reasoning trace. Instead, it compares positive and negative traces to construct a *relative preference signal*, which is more robust to noisy

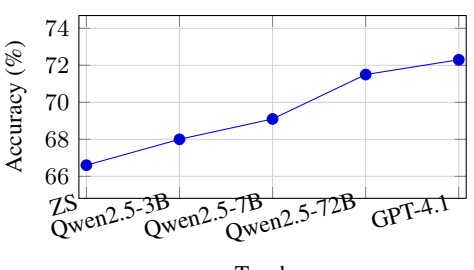

Figure 6: Teacher scaling: RAFT average accuracy improves monotonically with stronger teachers, validating robustness to teacher dependency.

or biased reasoning than absolute imitation. (2) **SFT as the dominant learning signal.** RAFT combines this reasoning-discriminative objective with standard SFT on ground-truth answers. The SFT loss anchors training to correct answer prediction, while the reasoning-discriminative term acts as a regularizer that nudges the model toward internally consistent reasoning without overpowering the final-task supervision. Together, these design choices ensure that RAFT leverages teacher signals while mitigating the risk of copying spurious reasoning patterns.

**Additional Experiments and Analysis.** We include efficiency analysis in Appendix A.4, results on text-only benchmarks in Appendix A.2, impact of incorrect reasoning supervision in Appendix A.5, impact on the predicted answer likelihood in Appendix A.6, ablation on the reasoning discriminative loss in Appendix A.7, training dynamic Appendix A.8 and comparison with DPO in Appendix A.9.

## 6 CONCLUSION

In this work, we introduced RAFT, a novel single-stage method designed to leverage a reasoning-discriminative loss to guide models toward preferring correct over incorrect reasoning steps without requiring reasoning generation during inference. This allows RAFT to retain the computational efficiency of SFT while improving input-output alignment and generalisation, particularly in low-data and reasoning-intensive settings. Extensive experiments on visual reasoning and fine-grained recognition benchmarks demonstrate that RAFT consistently outperforms SFT and CoT-based fine-tuning approaches across multiple shot settings. We believe RAFT offers a general and scalable solution for improving MLLMs under limited supervision and opens up new directions for reasoning-aware alignment without incurring inference-time costs.

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

## A  APPENDIX

### A.1  DATASETS AND BASELINES.

To comprehensively evaluate our method, we utilise a range of benchmark datasets spanning different visual tasks. For general visual reasoning, we employ the CVQA (Zhang et al., 2024a) dataset. To assess performance in the specialised domain of visual medical reasoning, we include OmniMed-VQA (Hu et al., 2024) and PMC-VQA (Zhang et al., 2023b). For this task, we used 128 samples for training and the remaining samples as the test set. Meanwhile, we also test the model's capabilities on fine-grained visual recognition in a challenging 1-shot, 4-shot, and 8-shot setting using five standard datasets: CUB (Caltech-UCSD Birds-200-2011 (Wah et al., 2011)), Stanford-Dogs (S.Dogs (Khosla et al., 2011)), Stanford-Cars (S.Cars (Krause et al., 2013)), FGVC-Aircraft (F.Aircraft (Maji et al., 2013)), and Oxford-IIIT Pet (O.Pet (Parkhi et al., 2012)). We split the training and test sets according to the number of shots.

To evaluate the performance of RAFT, we compare it against baselines representing different fine-tuning and evaluation strategies. These include: Zero-Shot (ZS) evaluation to assess the pre-trained model's capability without any fine-tuning; standard Supervised Fine-Tuning (SFT) on the target answers; Chain-of-Thought Fine-Tuning (COT-FT) (Kim et al., 2023; Muennighoff et al., 2025; Ye et al., 2025) where the model is trained on concatenated reasoning and answer sequences. We also compare our method DPO-like methods (DPO (Rafailov et al., 2023), SimPO (Meng et al., 2024), ORPO (Hong et al., 2024)).

### A.2  RESULTS ON TEXT-ONLY BENCHMARKS

To further validate the generality of RAFT beyond multimodal reasoning, we evaluate on text-only benchmarks covering commonsense reasoning and sentiment classification. Specifically, we consider **CommonsenseQA (CSQA)** and **IMDb sentiment classification**. The base model is Qwen2.5-7B (text-only variant).

Table 4: Performance comparison on text-only benchmarks. Scores are reported as accuracy (%). The base model is Qwen2.5-7B.

| Method | Text-only Benchmarks | |
|---|---|---|
| | CSQA | IMDb |
| SFT (Wei et al., 2021) | 73.2 | 91.5 |
| CoT-FT (Wei et al., 2022) | 74.1 | 91.0 |
| DPO (Rafailov et al., 2023) | 74.6 | 91.7 |
| RAFT (*Ours*) | **76.8** | **92.4** |

**Discussion.**  The results in Table 4 demonstrate that RAFT consistently outperforms both SFT and CoT-FT on text-only tasks. On CommonsenseQA, RAFT achieves a ∼2.7% absolute improvement over SFT, highlighting its ability to distill reasoning signals even in purely textual settings. On IMDb, RAFT yields a modest but consistent gain, showing that the reasoning-discriminative signal remains beneficial for classification-style tasks. These results confirm that RAFT's benefits are not confined to multimodal reasoning, but extend to general natural language understanding.

### A.3  ANALYSIS OF TEST SET LOG LIKELIHOOD.

Figure 7 illustrates the average log likelihood assigned to correct predictions on the test set, comparing the RAFT against the standard SFT baseline. The bar chart clearly indicates that the RAFT achieves an average log likelihood of approximately -0.014, whereas the SFT model attains a lower average log likelihood of approximately -0.019. A higher log likelihood (i.e., a less negative value) signifies that the model assigns greater

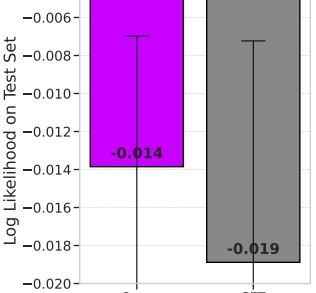

Figure 7: Log likelihood on test set.

likelihood to the ground-truth correct answers. This outcome suggests that RAFT develops a better internal model for generating correct outputs, leading to improved confidence and accuracy on unseen test data compared to SFT.

## A.4 EFFICIENCY ANALYSIS

To better understand the trade-offs of RAFT, we compare training and inference efficiency against SFT and CoT-FT on the **CUB-200-2011** dataset, using identical hardware (A100 80GB GPU) and training configurations. Table 5 reports the following metrics:

- **Annotation Time**: Average time to prepare one training example (reasoning traces required only for RAFT).
- **Training Time per Epoch**: Average time to complete one training epoch.
- **Training Throughput**: Number of training samples processed per second.
- **Average Output Length**: Average number of tokens generated during inference.
- **Inference Time**: Total time to process 1000 test samples.

Table 5: Comparison of training and inference efficiency across methods. RAFT introduces additional training overhead but retains SFT-level inference efficiency.

| Method | Annotation Time | Training Time / Epoch | Train Samples / Sec | Avg. Output Length | Inference Time |
|---|---|---|---|---|---|
| SFT | 0s | 1m 10s | 2.021 | 12 tokens | 17s |
| CoT-FT | 0s | 1m 15s | 1.869 | 210 tokens | 1m 25s |
| RAFT (*Ours*) | ∼0.065s | 4m 12s | 0.523 | 13 tokens | 17s |

**Observations.** We draw three key conclusions from Table 5:

1. **Training cost.** RAFT introduces overhead during training, with epoch time ∼3–4× longer than SFT and throughput reduced by more than half. This is primarily due to the reasoning-discriminative loss, which requires sampling and contrasting positive vs. negative reasoning traces.

2. **Annotation cost.** Preparing RAFT training instances adds a one-time annotation cost (∼0.065s per sample), which is amortized across epochs.

3. **Inference efficiency.** Despite higher training cost, RAFT produces short outputs comparable to SFT (12–13 tokens), in stark contrast to CoT-FT which generates ∼210 tokens. Consequently, RAFT matches SFT's inference speed (17s for 1000 samples) while being significantly faster than CoT-FT.

**Takeaway.** RAFT trades off higher *training* cost for improved reasoning-aware alignment, while retaining SFT-like *inference* efficiency. This makes RAFT particularly suitable for deployment scenarios where inference latency is critical but training resources are more flexible (e.g., server-side fine-tuning, low-shot domain adaptation).

## A.5 IMPACT OF INCORRECT REASONING SUPERVISION

To further underscore the critical importance of learning from *valid* reasoning pathways during training, we conducted an ablation study. In this experiment, instead of guiding the model with correct rationales as in RAFT, we explicitly prompted and trained the model to generate and internalise incorrect or flawed reasoning steps leading to (potentially) correct or incorrect final answers. The objective was to determine if the mere presence of a reasoning-like structure, even if faulty, could offer any benefit, or conversely, how detrimental learning incorrect reasoning would be. The performance impact was significant, as summarised below:

| Training Condition | Accuracy |
|---|---|
| Valid Reasoning Supervision | 65% |
| Incorrect Reasoning Supervision | 49% |

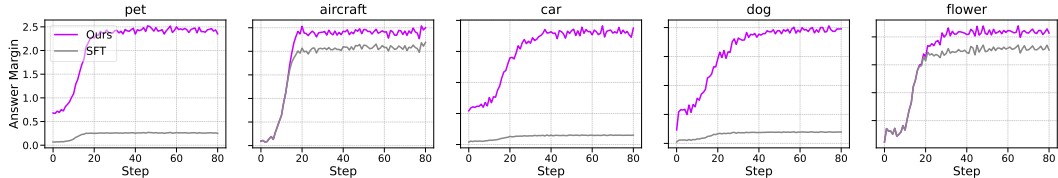

Figure 8: Comparison of the answer margins by RAFT (Ours) and SFT across training steps for various fine-grained visual recognition datasets: pet, aircraft, car, dog, and flower. This illustrates that reasoning and discrimination ability can encourage models to generate more confident answers.

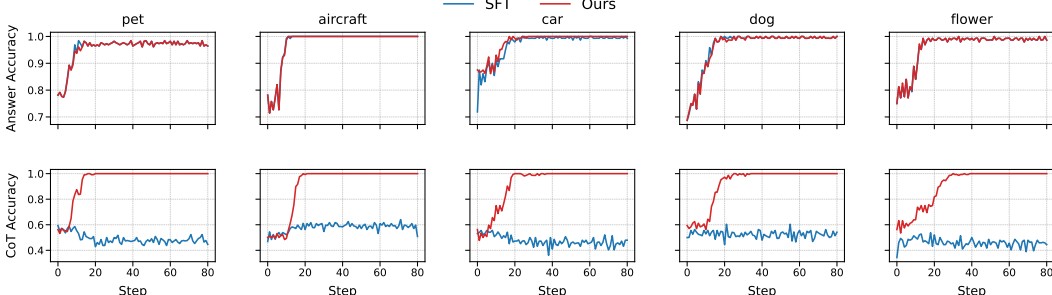

Figure 9: Training dynamics comparing RAFT (Ours) and SFT on fine-grained visual recognition datasets. The top row displays Answer Accuracy. The bottom row in the plot illustrates the model's ability to discriminate between positive and negative reasoning steps.

As evidenced by the table, compelling the model to learn from explicitly incorrect reasoning pathways led to a substantial degradation in task performance, with accuracy dropping from 0.65 to 0.49. This stark decrease highlights that the model does not simply ignore faulty reasoning but actively learns detrimental patterns, thereby corrupting its ability to arrive at correct solutions. This finding strongly supports our hypothesis that the positive impact of RAFT stems directly from the quality and correctness of the reasoning signals it integrates, and that indiscriminate or erroneous reasoning supervision is actively harmful.

### A.6 Impact on the Predicted Answer Likelihood

Figure 8 provides a comparative analysis of the answer margins achieved by RAFT versus standard SFT across various fine-grained visual recognition datasets, plotted over training steps. The "Answer Margin" on the y-axis represents the model's confidence in its prediction, the difference in log-probabilities between the chosen answer and the next best alternative. Consistently, across all five datasets, the RAFT model (labelled "Ours") exhibits a notably higher answer margin compared to SFT throughout the training process. This observation suggests that RAFT's enhanced reasoning discrimination capability, trained by distinguishing between positive and negative reasoning tokens, encourages the model to generate more confident and decisive correct answers.

### A.7 Ablation on the reasoning discriminative loss

As demonstrated in Table 6, incorporating the reasoning discriminative loss leads to a consistent performance improvement. The loss contains a term with a negative gradient that reduces the model's odds for the incorrect reasoning $z^-$. To assess its necessity, we conduct an ablation study where we removed this term. On the CUB fine-grained classification dataset, this leads to a performance drop of 2.9%. Furthermore, we investigate whether the gradient term for increasing the odds of the correct reasoning $z^+$ is equally crucial. Omitting this term resulted in a dramatic performance drop of 19%, underscoring its critical role in enabling the model to prefer high-quality reasoning. These results affirm that both positive and negative preference gradients are essential for RAFT's reasoning discrimination capabilities.

Table 6: Ablation of loss terms w.r.t relative accuracy.

| Method | Acc.(%) |
|---|---|
| RAFT | 71.50 |
| w/o $z^-$ | -2.90 |
| w/o $z^+$ | -19.00 |
| w/o $z^+$ & $z^-$ (SFT) | -4.90 |

## A.8 Training Dynamics

Figure 9 compares the training dynamics of RAFT and standard SFT across fine-grained visual recognition datasets. The top row illustrates *answer accuracy*, where RAFT (orange) consistently outperforms SFT (blue), achieving higher final accuracy across all datasets. Notably, the gap widens as training progresses, suggesting RAFT's reasoning-aware training enables more effective learning. The bottom row, labelled *CoT Accuracy*, measures the model's ability to discriminate between correct and incorrect reasoning steps by evaluating the margin between log-probabilities of positive versus negative reasoning tokens. RAFT exhibits a steady improvement in this metric, reflecting its enhanced internal reasoning discrimination. In contrast, SFT stagnates near chance levels ($50\%$), corroborating the hypothesis that SFT fails to internalise reasoning signals without explicit supervision. These results validate RAFT's dual benefit: improving answer accuracy while fostering robust reasoning discrimination, all without incurring the inference latency of explicit reasoning generation.

## A.9 Comparison between RAFT and DPO

Direct Preference Optimization (DPO) (Rafailov et al., 2023) is a preference-learning framework that optimizes a model to prefer responses $y^+$ over $y^-$ given the same prompt $x$, using a KL-regularised objective. Formally, DPO maximises:

$$\mathcal{L}_{\text{DPO}}(\theta) = \mathbb{E}_{(x,y^+,y^-)\sim\mathcal{D}} \left[ \log \sigma \Big( \beta \cdot \Big( \log \frac{\pi_\theta(y^+|x)}{\pi_{\text{ref}}(y^+|x)} - \log \frac{\pi_\theta(y^-|x)}{\pi_{\text{ref}}(y^-|x)} \Big) \Big) \right], \tag{11}$$

where $\pi_\theta$ is the student model, $\pi_{\text{ref}}$ is a frozen reference model, and $\beta$ controls the sharpness of the preference.

In contrast, RAFT introduces a *reasoning-discriminative* term over **reasoning traces**, not final outputs. Given a prompt $x$, ground-truth answer $y$, and teacher-sampled reasoning traces $z^+ = r^+ \oplus y$ and $z^- = r^- \oplus y^-$, RAFT minimizes:

$$\mathcal{L}_{\text{RAFT}}(\theta) = -\mathbb{E}_{(x,y,z^+,z^-)\sim\mathcal{D}_{\text{RAFT}}} \left[ \log \pi_\theta(y \mid x) + \beta \cdot \log \sigma \big( \text{logit}_\theta(z^+ \mid \tilde{x}) - \text{logit}_\theta(z^- \mid \tilde{x}) \big) \right], \tag{12}$$

where $\tilde{x}$ is a perturbed prompt used to decouple the SFT and reasoning objectives, and $\text{logit}_\theta(\cdot)$ denotes the normalized log-likelihood.

**Key Distinctions:**

- **Supervision signal:** DPO relies on *end-level preferences* over answers, while RAFT relies on *reasoning-level supervision* (contrasting valid vs. invalid reasoning traces).

- **Reference model:** DPO requires a frozen reference model $\pi_{\text{ref}}$ to anchor preferences. RAFT does not require a reference model, instead depending on a teacher model $\pi_T$ to generate contrastive reasoning.

- **Inference behavior:** DPO affects the model's distribution over final answers. RAFT preserves standard SFT decoding at inference and introduces reasoning supervision *only during training*, hence maintaining SFT-level inference efficiency.

- **Objective structure:** DPO is a KL-regularised preference optimisation objective; RAFT is a hybrid objective: SFT loss on final answers + reasoning-discriminative loss.

## A.10 Illustrative examples of Reasoning data generation

These examples demonstrate how both *positive* and *negative* reasoning traces are systematically derived from teacher model completions.

**Prompting the Teacher for Reasoning Traces.** To obtain reasoning–answer pairs from the teacher model $\pi^{\mathcal{T}}$, we use a structured prompt that explicitly separates the reasoning process from the final answer. The teacher is instructed to produce its chain of thought within special tags `<think>...</think>` and the final prediction within `<answer>...</answer>`. An example of the prompting template is shown below:

```
You are solving a visual question answering problem.

**Action: Thinking**
- Outline the step-by-step thinking process to solve the problem.
- Use the <think> tags to detail your process.

**Action: Answer**
- Output your final answer within the <answer> tag with
just one word or one option.

Example:
<answer>5</answer>

Q: {question}
A:
```

Given this query format, each sampled completion naturally decomposes into two parts: (1) a reasoning trace $r_i$ extracted from the `<think>` block, and (2) a final answer $y_i$ extracted from the `<answer>` block. We then categorize the trace as a **positive sample** $z_i^+ = r_i^+ \oplus y_i^+$ if $y_i^+ = y_i$ matches the ground-truth label, or as a **negative sample** $z_i^- = r_i^- \oplus y_i^-$ if $y_i^- \neq y_i$.

This explicit prompting strategy guarantees a clean separation between reasoning and answers, enabling us to systematically construct contrastive reasoning pairs. Moreover, it ensures reproducibility: the same template is applied consistently across all benchmarks.

**Extracting Reasoning Traces and Answers.** Given a prompt $x_i$, we query the teacher model $\pi^{\mathcal{T}}$ using a structured template that explicitly separates reasoning from the final prediction. The teacher is instructed to output their reasoning process inside `<think>` tags, and the final answer inside `<answer>` tags. This makes it straightforward to parse the output into a reasoning trace $r_i$ and a candidate answer $y_i$.

For example, the teacher may produce:

```
<think>
The bird has bright yellow underparts and a thin curved beak,
traits characteristic of a yellow warbler.
</think>
<answer>Yellow Warbler</answer>
```

From this output, we extract:
$$z_i^+ = r_i^+ \oplus y_i^+ \quad \text{if } y_i^+ = y_i,$$
where $y_i$ is the ground-truth label. If instead the final answer $y_i^-$ does not match $y_i$, the corresponding reasoning trace $z_i^- = r_i^- \oplus y_i^-$ is treated as a negative example, provided that the reasoning remains semantically plausible.

**Implementation detail.** We use a lightweight regular expression to extract the answer token from the teacher's output:

```
match = re.search(r'<answer>(.*?)</answer>', output, re.DOTALL)
```

This guarantees consistent answer parsing across all datasets. Reasoning traces are preserved from the `<think>` block, while only the final token inside `<answer>` is compared against the ground truth.

In the first example (fine-grained bird classification), the teacher produces multiple reasoning–answer completions for the input image prompt "What is the type of this bird?". From these, we select a **positive trace** that ends with the correct answer (*Yellow Warbler*) and whose intermediate reasoning is logically consistent (e.g., identifying yellow underparts and a thin curved beak). We then select a **negative trace** that ends with an incorrect label (*Song Thrush*), but whose reasoning remains plausible (e.g., citing mottled brown plumage and a stout beak). This ensures that the negative is not trivial noise but a semantically coherent alternative.

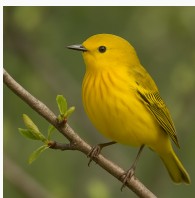

- Prompt with image ($\mathbf{x}_i$): "What is the type of this bird?"
- Positive Response ($\mathbf{r}_i^+ \oplus \mathbf{y}_i^+$): "It has bright yellow underparts and a thin, slightly curved beak—features characteristic of a **yellow warbler**."
- Positive Answer ($\mathbf{y}_i^+$): "Yellow Warbler"
- Negative Response ($\mathbf{r}_i^- \oplus \mathbf{y}_i^-$): "It has mottled brown plumage and a stout beak—traits seen in many thrush species. So it is a **Song Thrush**."
- Negative Answer ($\mathbf{y}_i^-$): "Song Thrush"

In the second example (CommonsenseQA), the prompt "Where would you expect to find a pizzeria while shopping?" with multiple-choice answers is used. The **positive trace** correctly reasons that shopping malls contain food courts with restaurants, leading to the answer *food court*. The **negative trace**, in contrast, incorrectly concludes that because Chicago is famous for deep-dish pizza, the answer should be *chicago*. Here again, the negative remains contextually relevant but incorrect, providing a meaningful contrastive supervision signal.

> **Question:** Where would you expect to find a pizzeria while shopping?
> A: chicago    B: street    C: little italy    D: food court    E: capital cities
> **Positive CoT (chosen):** *When people go shopping, especially in malls or shopping centers, there are usually food courts where a variety of restaurants, including pizzerias, are located. Therefore, the answer is food court.*
> **Negative CoT (rejected):** *Chicago is famous for deep-dish pizza, so one might expect to find a pizzeria there. Therefore, the answer is chicago.*

These examples illustrate how RAFT leverages contrastive reasoning supervision: for each instance, we guarantee at least one valid positive and one valid negative reasoning trace, ensuring stability during training. By pairing logically sound but incorrect reasoning with correct reasoning, RAFT teaches the student model to discriminate between valid and invalid reasoning paths without requiring explicit reasoning generation at inference time.

## A.11 REPRODUCIBILITY STATEMENT

Code, training scripts, and evaluation utilities will be publicly released upon publication.

## A.12 AI USAGE CLARIFICATION

Large Language Models were employed solely to enhance grammar and readability. All aspects of research design, analysis, and interpretation were conducted by the authors.

