# OpenReview forum: "RAFT: Reasoning-Aware Fine-Tuning"
_ICLR.cc/2026/Conference — ICLR 2026 Conference Withdrawn Submission_

### Official Review · Reviewer_WFoZ · 2025-10-30

**Soundness:** 3
**Presentation:** 3
**Contribution:** 2
**Rating:** 4
**Confidence:** 3

**Summary:**

The authors present a new SFT method that aims to address an issue of the classic SFT approach, which causes the model to build only query-output mapping, which sometimes is not consistent with the reasoning traces. The proposed RAFT method includes preferences on correct reasoning traces, strengthening the base models’ ability to identify both the reasoning trace and final answer correctly.

After reading the paper, I’m under the impression that the proposed method in the paper is conceptually sound, and the experimental setup is solid, well supporting the main claims. However, I’m not fully convinced that the proposed method is indeed significant and novel in terms of eliciting the reasoning capabilities of LLM, compared to common SFT + RL strategies used nowadays. I’m willing to engage in more discussion with the authors to get more insights on the RAFT method.

**Strengths:**

1. The authors identifies a core problem in SFT, which is the gap between reasoning and shallow input-output mapping, which impedes the generalization and reasoning performance of the model. The proposed method is well motivated and successfully addresses the problem, evaluated on some empirical settings.
2. The paper provides strong experimental results, supporting the main claims. Sufficient ablation studies are conducted, making the proposed method more robust and plausible.
3. The proposed RAFT method has efficiency advantages, improving the token efficiency compared to other reasoning-based SFT method (e.g. CoT-FT), which potentially makes it applicable and scalable.

**Weaknesses:**

1. The experimental results are limited to mostly visual reasoning, lacking some results in more diverse settings, such as mathematical reasoning.
2. The essential concept of the proposed method doesn’t seem to differ much from preference optimization algorithms, which limits the potential impact of the work, and the novelty of the method is limited.
3. The proposed method seems to be limited by the quality of the teacher model, that generates the reasoning traces. This could limit the scaling of the algorithm
4. No supplementary materials or code are provided. I encourage the authors to open-source the code for better transparency and reproducibility.
5. Missing discussions on recent reasoning-focused SFT methods [1, 2]

[1] s1: Simple test-time scaling (https://arxiv.org/abs/2501.19393)
[2] LIFT the Veil for the Truth: Principal Weights Emerge after Rank Reduction for Reasoning-Focused Supervised Fine-Tuning (https://arxiv.org/abs/2506.00772)

**Questions:**

1. For the RAFT objective, since the model wasn’t rained on reasoning traces, I’m wondering whether the Reasoning Discrimination loss will be very small (because of the probability of predicting $\mathbf{z}^+$ and $\mathbf{z}^-$ will both be small), causing the signal from the reasoning discriminative loss to be very weak? This might not help the model to truly discriminate between positive and negative reasoning traces.
2. Given that there might be imbalance between the SFT loss and Reasoning Discriminative loss, the optimization of two losses may happen in different stages of the training. Could the authors provide the training curve for both losses? It might give us better insights into the training dynamics of the two components.
3. The experimental results provided in the paper are all vision-language results. It would be better if the authors could provide some additional results on language-based reasoning, such as mathematical reasoning and commonsense reasoning. Also it would benefit more if the authors could compare the RAFT method with the most commonly used RL methods such as DAPO, GRPO.

---

### Official Review · Reviewer_ab9x · 2025-10-31

**Soundness:** 3
**Presentation:** 3
**Contribution:** 2
**Rating:** 4
**Confidence:** 3

**Summary:**

This paper proposes a new SFT loss that prefers correct reasoning traces over incorrect ones. Through experiments on both vision-language and text-only tasks, the authors demonstrate that the proposed method, RAFT, outperforms direct-answer SFT, CoT-SFT, and DPO.

**Strengths:**

1. The proposed method is clearly described.
2. The proposed method is simple yet effective, demonstrating strong performance on multi-modal tasks compared with a variety of existing methods.

**Weaknesses:**

1. The method is relatively incremental, adding an additional contrastive term to the SFT loss, which is similar to the Iterative Reasoning Preference Optimization approach [1].
2. The evaluation datasets are quite limited, mainly focusing on tasks where CoT performance is weak, especially since CoT-SFT performs worse than direct-answer SFT. This may give RAFT an unfair advantage.
3. The comparison includes only a small number of models. Each of the visual and text tasks uses just one policy model, so it remains unclear whether the method generalizes to other model sizes or families.

[1] Iterative Reasoning Preference Optimization, https://arxiv.org/pdf/2405.14734.

**Questions:**

Corresponding to the weaknesses:

1. Could you compare your method with Iterative RPO (which extends the DPO loss with an NLL term)?
2. Could you provide the CoT zero-shot results, especially on fine-grained visual recognition benchmarks?
3. Does CoT-FT use CoTs derived from the correct reasoning traces of the teacher model, or from a different source? Then, could you explain why CoT-FT performs worse than SFT, for example, is it due to the quality of the CoTs?
4. Could you also provide results using other models from different model families?
5. Is it possible to generalize RAFT to commonly used datasets like Math?

---

### Official Review · Reviewer_P21g · 2025-11-01

**Soundness:** 2
**Presentation:** 3
**Contribution:** 2
**Rating:** 4
**Confidence:** 3

**Summary:**

The paper introduces RAFT (Reasoning-Aware Fine-Tuning), a single-stage fine-tuning framework that integrates a reasoning-discriminative loss to teach models to prefer valid reasoning over incorrect ones without requiring reasoning generation at inference.

**Strengths:**

(1) The proposed reasoning-aware fine-tuning seems novel.


(2) The paper is well-organized and clearly written.

(3) Extensive experiments demonstrate the effectiveness of the proposed RAFT Method.

**Weaknesses:**

(1) Lack of baselines from RL fine-tuning. Reinforcement learning (RL) fine-tuning methods such as GRPO and Dr.GRPO [1,2] have recently shown strong improvements in reasoning ability. Including these approaches as baselines is essential for a fair and comprehensive evaluation of the proposed method’s contributions.

(2) Unconvincing baseline performance of COT-FT. In Figure 2, the COT-FT model performs significantly worse than the SFT baseline, which contradicts the findings reported in [3,4]. The authors should provide an explanation or verification to clarify this discrepancy.

(3) Minor presentation issues. The tick labels in Figure 5 are too small and not clearly visible. Moreover, the formatting is inconsistent with Figure 6.


[1] Shao, Zhihong, et al. "Deepseekmath: Pushing the limits of mathematical reasoning in open language models." arXiv preprint arXiv:2402.03300 (2024).

[2] Liu, Zichen, et al. "Understanding r1-zero-like training: A critical perspective." arXiv preprint arXiv:2503.20783 (2025).

[3] Yixin Ye, Zhen Huang, Yang Xiao, Ethan Chern, Shijie Xia, and Pengfei Liu. Limo: Less is more for reasoning. arXiv preprint arXiv:2502.03387, 2025.

[4] Seungone Kim, Se June Joo, Doyoung Kim, Joel Jang, Seonghyeon Ye, Jamin Shin, and Minjoon Seo. The cot collection: Improving zero-shot and few-shot learning of language models via chain-of-thought fine-tuning. arXiv preprint arXiv:2305.14045, 2023.

**Questions:**

(1) What's advantages of the proposed RAFT VS. RL-based method such as GRPO/Dr. GRPO? Can RAFT outperform GRPO-based methods?

(2) Is there any explanation why COT-FT performens much worse than FT in Figure 2.

---

### Official Review · Reviewer_2p8C · 2025-11-02

**Soundness:** 2
**Presentation:** 3
**Contribution:** 2
**Rating:** 4
**Confidence:** 3

**Summary:**

The author proposed a fine-tuning pipeline named RAFT to enhance the efficiency of reasoning. The pipeline is designed following the paradigm of contrastive learning to train the student network to produce both the expected reasoning and the answer. Through extensive experiments, the authors demonstrate that RAFT achieves better performance compared to other fine-tuning methods.

**Strengths:**

1.The design of the pipeline is more effective and accurate in finding the right ways to right answers.
2.The entire method is low-invasive and can be well integrated into most of the training process. The RAFT has shown promising results on different areas like vision reasoning and medical VQA.

**Weaknesses:**

1. The authors claimed that a large beta might overshadow the primary task objective, leading to a performance drop. This behavior is potentially problematic for the motivation of RAFT. Intuitively, increasing beta should strengthen the model’s ability to learn correct reasoning and thereby increase the accuracy of final results. However, larger beta values (e.g., 1 or 0.1) yield higher CoT accuracy but lower reported final answer accuracy. This means that there could exist a lack of connection between the CoT and the answer.

2. The data in the article may contain errors. In Table 1, the performance of RAFT are 64.47/87.63/53.53, while in the explanation texts (line 375), they become 65.08/88.75/53.60.

3. The authors partially reported the variances of RAFT in Table 1 & 2, but other tables/tasks have fewer significance tests and confidence intervals, which lacks uniformity.

**Questions:**

How does the reported behavior of β align with your stated motivation? If there is a mismatch, please explain it and provide supporting evidence.

There appear to be numerical inconsistencies (Table 1 vs line 375) and uneven statistical reporting; which numbers are correct, and can you standardize the reporting (seeds, variance) to support your claims?

---

### Note · Authors · 2025-12-03

I have read and agree with the venue's withdrawal policy on behalf of myself and my co-authors.